# Genome Size, Flowering, and Breeding Compatibility in Osmanthus Accessions

Lisa Alexander 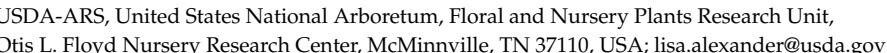

USDA-ARS, United States National Arboretum, Floral and Nursery Plants Research Unit,
Otis L. Floyd Nursery Research Center, McMinnville, TN 37110, USA; lisa.alexander@usda.gov

**Abstract:** Extending the range of *Osmanthus* species into more cold-hardy climates would open new opportunities for adoption and use of these species by growers, landscapers, and the public. Breeding improvement is hindered by few available cultivars and a lack of female or perfect flowers. The objectives of this study were to evaluate floral morphology and pollination biology of *Osmanthus* species available in the U.S. market. Thirty-three genotypes representing four species were evaluated in McMinnville, TN, USA for genome size, floral morphology, pollen viability, and self- and cross-compatibility. All genotypes were diploid with 2C genome sizes ranging from 2.89 to 3.21 pg. Perfect flowers were observed in all *O. armatus* and 82% of *O. heterophyllus* genotypes. All observed *O. fortunei* and *O. fragrans* genotypes had male-only flowers. Pollen viability based on staining and in vitro germination ranged from 8% to 98% and 6% to 53%, respectively. Pollen germination was observed on stigmas of 94.2% of *O. armatus* and *O. heterophyllus* flowers collected 24 hours after cross- or self-pollination. There was a significant association between cross type and percentage of flowers with pollen tubes reaching the ovaries; after 72 h, pollen tubes had reached the ovaries of in 67% of intraspecific crosses, 78% of interspecific crosses, and 0% of self crosses ($\chi^2 = 26.5$, $p < 0.001$). This study provides evidence of a self-incompatibility system in *O. armatus* and *O. heterophyllus* and provides insights into opportunities and challenges for *Osmanthus* hybrid breeding.

**Keywords:** pollen tube growth; breeding system; self-incompatibility; tea olive; sweet olive





## 1. Introduction

Ornamental flowering trees enjoy worldwide popularity for their impact in the landscape [1]. The genus *Osmanthus*, in the Olive family (Oleaceae), consists of about 30 species of evergreen trees and shrubs distributed primarily throughout temperate and tropical China [2]. Several species have cultivars available in the U.S. market including *Osmanthus fragrans*, *O. heterophyllus*, and *O. armatus* [3]. A well-known interspecific *Osmanthus* hybrid, *O. heterophyllus* x *O. fragrans*, is named *O. fortunei*, or Fortune's osmanthus, for famous plantsman Robert Fortune. Fortune introduced this plant to England from Japan by way of Holland in 1862 [3]. The hybrid was developed at least twice more, by the W.B. Clarke nursery of San Jose, California and Fruitland Nurseries in Augusta, Georgia. Both 'San Jose' and 'Fruitlandii' have fragrant flowers and are more cold-tolerant than *O. fragrans* [3,4].

Despite these promising moves within breeding improvement of *Osmanthus*, areas colder than USDA Hardiness Zone 7 lack cultivars with consistent growth, flowering, and *O. fragrans* form [3]. Extending the range of *Osmanthus* species into more cold-hardy climates throughout both the U.S. and China would open new opportunities for adoption and use of these species by growers, landscapers, and the public [5,6]. Production and landscape use is currently limited to USDA hardiness zones 7–10, but there is a wide range in cold-hardiness among species and cultivars [3,5]. *Osmanthus heterophyllus* and its hybrid *O. fortunei* were the most cold-hardy while *O. fragrans* was the least cold-hardy in three-year evaluation of growth, winter damage, and flowering in McMinnville, TN,

USA [4]. Based on the intermediate cold-hardiness of the *O. heterophyllus × O. fragrans* hybrid and the presence of mature landscape plants *O. heterophyllus* observed as far north as USDA hardiness zone 6b [3], there is strong potential for breeding improvement to deploy these species into wider areas.

The unique reproductive systems found across plant species play a major role in the techniques, timing, and strategies of plant breeding efforts [7]. Approximately 40% of angiosperms, including many woody trees and shrubs, have mechanisms which limit or prevent self-fertilization [8]. Self-incompatibility (SI) refers to a collection of diverse mechanisms used by plants to reject self-pollen or pollen from genetically related individuals to prevent inbreeding or promote outcrossing [9–11]. Self-incompatibility may result from a single multi-allelic gene or several genes [11]. In the two most common systems, SI is regulated by either the haploid genotype of the pollen grain (gametophytic incompatibility) or the diploid genotype of the pollen parent (sporophytic incompatibility) [9–11]. Because individual plants can possess identical alleles at SI loci, SI systems can have consequences beyond self-pollination. SI systems can limit compatible matings among genotypes, limiting production potential where seed set is important and limiting successful crosses in breeding programs [7,12].

While the SI system of *Osmanthus* has not been documented, recent breeding studies in olive (*Olea europaea*) have uncovered a sporophytic SI system controlled by a single gene that regulates a series of cellular interactions at the stigmatic surface of the pistil [13]. In this system, SI is manifested by the inhibition of pollen hydration, pollen tube germination, or pollen tube infiltration of the stigmatic surface [7,13]. All olive cultivars examined are self-incompatible, though "leaky" SI has been reported in a few genotypes in the presence of large quantities of incompatible pollen [7]. In order to understand the pre and post zygotic reproductive barriers at work in a species, pollen germination and pollen tube growth can be visualized using microscopy. The type of reproductive barriers in place influence the choice of plant breeding techniques that may be used to overcome SI [14].

*Osmanthus* flowers can be male-only or perfect (i.e., hermaphrodite, possessing both male and female floral organs), a breeding system known as androdioecy [15,16]. *Osmanthus fragrans* has a 1:1 (perfect: male-only) sex ratio in natural populations across its native range in China which has been interpreted as a snapshot in time on the evolutionary path to dioecy, where male flowers and female flowers exist on separate plants [16]. All of the *O. fragrans* cultivars commonly available on the U.S. market appear to be male-only, and other *Osmanthus* species may have some cultivars with male-only flowers and some with perfect flowers. These variations in flowering time and flower function are obstacles to hybridization among *Osmanthus* species. Polyploidy—or whole genome duplication—can also serve as a reproductive barrier among otherwise compatible species, as gametes with uneven numbers of chromosome fail to unite at fertilization [17]. Almost 70% of flowering plants have undergone ploidy changes in their evolutionary history [18], such that ploidy complexes can be major barriers for breeding compatibility [17,19].

A comprehensive evaluation of flowering and floral biology is needed to enable breeding improvement in *Osmanthus*, both to create novel genetic variation and introgress favorable traits from *O. fragrans* into more cold-hardy *Osmanthus* species. The objectives of this study were to evaluate genome size, floral morphology, pollen viability, and self- and cross-compatibility of *Osmanthus*. Results of this study will enable breeders to identify parental pair candidates, guide future collection efforts, and develop breeding and evaluation strategies for *Osmanthus*.

## 2. Materials and Methods

### 2.1. Plant Materials

Twenty-six *Osmanthus* cultivars and seven unnamed accessions representing four species (n = 33 genotypes) were obtained from the U.S. National Arboretum or commercial nurseries and grown in containers at the Tennessee State University Nursery Research Center in McMinnville, TN, USA (Table 1). Plants were maintained in 7-gallon pots under full sun

and micro-irrigated using spray stakes. Growing media consisted of pine bark amended with 6.6 kg·m$^{-3}$ 19N-2.1P-7.4K Osmocote Pro fertilizer (Scotts-Sierra Horticultural Products Co., Maryville, OH, USA), 0.6 kg·m$^{-3}$ Micromax (Scotts-Sierra Horticultural Products Co.), 0.6 kg·m$^{-3}$ iron sulfate, and 0.2 kg·m$^{-3}$ Epsom salts. Pots were moved into a plastic covered polyhouse heated to 6 C from 21 November 2017 through 1 May 2018.

**Table 1.** Thirty-three accessions representing four species of *Osmanthus* used for genome sizing, floral morphology, pollen viability, and/or incompatibility studies at the U. S. National Arboretum in McMinnville, TN, USA. A dash (-) indicates seedling (i.e., not a named cultivar).

| Genus | Species | Cultivar | Immediate Source | Original Source |
|---|---|---|---|---|
| *Osmanthus* | *armatus* | - | USNA | Woodlanders |
| *Osmanthus* | *armatus* | Jim Porter | Nurseries Caroliniana | Nurseries Caroliniana |
| *Osmanthus* | *armatus* | Longwood | Nurseries Caroliniana | Nurseries Caroliniana |
| *Osmanthus* | *fortunei* | - | Nurseries Caroliniana | Nurseries Caroliniana |
| *Osmanthus* | *fortunei* | - | USNA | USNA S. Japan 1976 |
| *Osmanthus* | *fortunei* | Fruitlandii | Nurseries Caroliniana | Nurseries Caroliniana |
| *Osmanthus* | *fortunei* | San Jose | Nurseries Caroliniana | Nurseries Caroliniana |
| *Osmanthus* | *fragrans* | - | USNA | Mrs. Charles Truitt |
| *Osmanthus* | *fragrans* | - | USNA | Woodlanders |
| *Osmanthus* | *fragrans* | - | USNA | Kobayashi Nursery |
| *Osmanthus* | *fragrans* | Apricot Echo | Nurseries Caroliniana | Nurseries Caroliniana |
| *Osmanthus* | *fragrans* | Beni Kin Mokusei | Nurseries Caroliniana | Nurseries Caroliniana |
| *Osmanthus* | *fragrans* | Daye Fondingzhu | Nurseries Caroliniana | Nurseries Caroliniana |
| *Osmanthus* | *fragrans* | Fudingzhu | Nurseries Caroliniana | Nurseries Caroliniana |
| *Osmanthus* | *fragrans* | Quinnan Guifei | Nurseries Caroliniana | Nurseries Caroliniana |
| *Osmanthus* | *fragrans* | Thunbergii | Nurseries Caroliniana | Nurseries Caroliniana |
| *Osmanthus* | *fragrans* | Thunbergii Clemson Hardy | Nurseries Caroliniana | Nurseries Caroliniana |
| *Osmanthus* | *fragrans* | Tianxiang Taige | Nurseries Caroliniana | Nurseries Caroliniana |
| *Osmanthus* | *fragrans* | Yinbi Shuanghui | Nurseries Caroliniana | Nurseries Caroliniana |
| *Osmanthus* | *heterophyllus* | - | USNA | Natl. Park Serv. Nursery |
| *Osmanthus* | *heterophyllus* | Goshiki | Nurseries Caroliniana | Nurseries Caroliniana |
| *Osmanthus* | *heterophyllus* | Hariyama | Nurseries Caroliniana | Nurseries Caroliniana |
| *Osmanthus* | *heterophyllus* | Head-Lee Fastigate | Nurseries Caroliniana | Nurseries Caroliniana |
| *Osmanthus* | *heterophyllus* | Illicifolius | USNA | Unknown |
| *Osmanthus* | *heterophyllus* | Kaori Hime | Nurseries Caroliniana | Nurseries Caroliniana |
| *Osmanthus* | *heterophyllus* | Kembu | USNA | Suzuki Nursery |
| *Osmanthus* | *heterophyllus* | Ogon | Nurseries Caroliniana | Nurseries Caroliniana |
| *Osmanthus* | *heterophyllus* | Purpureus | USNA | Gossler Farms |
| *Osmanthus* | *heterophyllus* | Rotundifolius | Nurseries Caroliniana | Nurseries Caroliniana |
| *Osmanthus* | *heterophyllus* | Rotundifolius Nouvea | Nurseries Caroliniana | Nurseries Caroliniana |
| *Osmanthus* | *heterophyllus* | Sasaba | USNA | Woodlanders |
| *Osmanthus* | *heterophyllus* | Shien | Nurseries Caroliniana | Nurseries Caroliniana |
| *Osmanthus* | *heterophyllus* | Variegatus | Nurseries Caroliniana | Nurseries Caroliniana |

### 2.2. Genome Sizing

Twenty-seven genotypes representing four species were available for genome size determination: *O. armatus* (n = 3 genotypes), *O. fortunei* (n = 4 genotypes), *O. fragrans* (n = 7 genotypes), and *O. heterophyllus* (n = 13 genotypes). Approximately 0.5 cm$^2$ of growing leaf tissue of sample and standard were chopped for 30 to 60 s in a plastic Petri dish containing 0.4 mL extraction buffer (Partec CyStain ultraviolet precise P Nuclei Extraction Buffer; Partec GMBH, Muenster, Germany). The resulting extract was passed through a 30 μM filter into a 3.5-mL plastic tube to which was added 1.6 mL Partec CyStain ultraviolet precise P Staining Buffer containing the fluorochrome 4′, 6-diamidino-2-phenylindole (DAPI). The relative fluorescence of the total DNA was measured for each nucleus using a Partec PA-1 ploidy analyzer (Partec GMBH, Muenster, Germany). Results were displayed as histograms showing the number of nuclei grouped in peaks of relative fluorescence intensity. For each sample, at least 3000 nuclei were analyzed revealing a single peak with a coefficient of variation (CV) less than 4.9%. Genome sizes were calculated as nuclear DNA content for unreduced tissue (2C) as: 2C DNA content of tissue = (mean fluorescence value of sample ÷ mean fluorescence value of standard) × 2C DNA content of standard. *Zea maize* L. 'Ctirad' with a 2C content of 5.67 pg was used as the internal standard [20]. Ploidy and genome sizes are the averages of four samples per genotype (two subsamples per plant; two plants [clones] per genotype).

### 2.3. Floral Morphology

Three clones per accession were examined for flowers between 8 September 2017 and 17 May 2018. Presence or absence of flowers was recorded for each plant. Where present, flowers were collected and observed under a light microscope with 2× magnification. Type of flower (male, female, or perfect), floral organs (normal or reduced), and presence/absence of pollen was recorded.

### 2.4. Pollen Staining and In Vitro Germination

To estimate pollen viability, fresh pollen from a single flower was placed on a microscope slide using a camel-hair brush (n = 3 flowers per plant). A 30 μL drop of modified Alexander's stain [21] was pipetted onto the slide and a coverslip was applied. Slides were observed after 30 minutes at 10× magnification using an Olympus BX-60 compound microscope with an Olympus Q Color 5 digital camera for image capture. Pollen was counted in three fields of view containing at least 50 pollen grains. Pollen was sampled three times: 13–14 November 2017, 7–8 December 2017, and 9 January 2018 for a total of 9 flowers per genotype (three flowers at three time points). Genotypes were arranged in a completely randomized design.

Pollen viability was further examined through in vitro germination. Pollen germination media (PGM) contained 7.5% sucrose, 2-(N-orpholino)ethanesulfonic acid (MES; 10 mM) pH 5.5, 0.01% CaNo3, 0.01% H3BO3, and 0.01% KNO3 in 0.5% agar substrate. 5 mL of media was pipetted into each 60 mm Falcon plastic dish (Corning Inc., Corning, NY, USA). For each accession, anthers from ten flowers were removed and placed in a glass jar. A camel-hair brush was dipped in the pollen and lightly dragged across the surface of the agar-based germination media. New brushes were used each time. Petri dishes were covered and inverted immediately after pollen was applied. After 20 h at room temperature, pollen was visualized at 10× magnification with an Olympus BX50 compound microscope (Olympus Corp., Tokyo, Japan) with an Olympus Q Color 5 digital camera for image capture. Measurements and counts were made using Q-Capture Pro 7 software (Q Imaging 2010). Pollen grains were classified into one of three categories: oblong, round/ungerminated, or round/germinated. Pollen of each category in a single field of view with at least 100 total pollen grains were counted and recorded. Three replicates (dishes) per accession were used in tests of pollen germination. Replicates were arranged in a completely randomized design.

### 2.5. Controlled Pollinations

Six genotypes were used in controlled pollinations: *Osmanthus armatus* 'Jim Porter' and 'Longwood'; *O. fragrans* 'Fudingzhu'; and *O. heterophyllus* 'Kaori Hime', 'Rotundifolius', and 'Shien'. Pollen receptor flowers were emasculated 1 day before pollination and covered with breathable bags (DelStar, Inc., Middleton, DE, USA). Pollen from donor flowers was collected in glass vials the day of pollination, pollen was applied to exposed receptor stigmas using a camel hair brush, and the bag was replaced. A new brush was used for each pollination. Bags were removed after 1–2 weeks.

To analyze pollen tube growth, approximately 15 flowers were pollinated for each cross as above. Flowers were collected 24, 48, or 72 h after pollination and prepared for visualization. Briefly, pistils were removed from flowers and fixed in Carnoy I (3:1; ethanol:acetic acid) for at least 24 h at room temperature. The fixative was removed by pipetting and pistils were rinsed twice for 10 min in 70% ethanol with gentle agitation. Fresh 70% ethanol was added to cover pistils and pistils remained in ethanol until visualization (between 24 h and 6 d). On the day of visualization, ethanol was removed and pistils were rinsed in deionized water, hydrolyzed in 8 N NaOH for 1–3 h, and rinsed again with deionized water. Fully hydrolyzed pistils appeared translucent. Pistils were placed in a Petri dish containing decolorized aniline blue (0.1% aniline blue dissolved in 0.1 N K3PO4) for 1 h, removed to a microscope slide with one 30 μL drop of stain (2:1; decolorized aniline blue:glycerin), and gently flattened with a coverslip. Slides containing stained pistils were visualized using a fluorescent microscope (BX-60; Olympus America, Inc., Center Valley, PA, USA) equipped with a 100 W high-pressure Hg lamp and a U-MNV near ultraviolet (400–410 nm) filter at $4\times$ magnification. After slide preparation, between 3 and 6 intact flowers were available for each cross at each time point. Data recorded for three to five flowers per cross included length of longest three pollen tubes (mm) and whether pollen tubes reached the ovary (Y/N). Total length of each flower was recorded. A total of 142 pollinated flowers were visualized, measured, and used for further analysis.

### 2.6. Data Analysis

Pollen viability based on staining was calculated as: % round, stained grains = (number round, stained grains $\div$ total number of grains) $\times$ 100%. Pollen viability based on in vitro germination was calculated as: % round, germinated grains = (number round, germinated grains $\div$ total number of grains) $\times$ 100%. One-way ANOVA mixed models were used to partition variance in genome size, pollen viability, and pollen germination into sources attributable to species, cultivar, and environment (error). Pearson's chi-square test of association was used to determine the relationship between cross type and pollen tube growth. Analysis of covariance was used to partition variance in pollen tube length into sources attributable to cross type and environment (error) with female flower length as a covariate. Means for each species, cultivar, or cross type were compared using Tukey's studentized range test with an $\alpha = 0.05$ significance level. All data analysis was performed using SAS® software, Version 9.4 of the SAS system for Microsoft (Copyright © 2013, SAS Institute Inc., Cary, NC, USA).

## 3. Results

### 3.1. Genome Size

All *Osmanthus* species and cultivars (n = 27) were diploid, with 2C genome sizes ranging from 2.89 pg for *O. fragrans* 'Beni Kin Mokusei' to 3.21 pg for *O. heterophyllus* 'Hariayama' (Table 2). Species was a significant source of variation in genome size (F = 11.5; $p < 0.001$). The *O. fragrans* genome ($2.94 \pm 0.04$ pg) was significantly smaller than that of *O. heterophyllus* ($3.10 \pm 0.06$ pg), (Table 3).

**Table 2.** Genome size of twenty-seven *Osmanthus* accessions. A dash (-) represents an unnamed seedling.

| Genus | Species | Cultivar | Genome size (pg) (Mean ± Standard Deviation) |
|---|---|---|---|
| *Osmanthus* | *armatus* | - | 3.03 ± 0.02 |
| *Osmanthus* | *armatus* | Jim Porter | 3.09 ± 0.09 |
| *Osmanthus* | *armatus* | Longwood | 3.08 ± 0.03 |
| *Osmanthus* | *fortunei* | - | 3.04 ± 0.09 |
| *Osmanthus* | *fortunei* | - | 2.96 ± 0.06 |
| *Osmanthus* | *fortunei* | Fruitlandii | 3.03 ± 0.10 |
| *Osmanthus* | *fortunei* | San Jose | 3.02 ± 0.04 |
| *Osmanthus* | *fragrans* | - | 3.02 ± 0.01 |
| *Osmanthus* | *fragrans* | - | 2.92 ± 0.03 |
| *Osmanthus* | *fragrans* | Apricot Echo | 2.91 ± 0.02 |
| *Osmanthus* | *fragrans* | Beni Kin Mokusei | 2.89 ± 0.02 |
| *Osmanthus* | *fragrans* | Fudingzhu | 2.94 ± 0.07 |
| *Osmanthus* | *fragrans* | Thunbergii | 2.98 ± 0.10 |
| *Osmanthus* | *fragrans* | Thunbergii Clemson Hardy | 2.95 ± 0.07 |
| *Osmanthus* | *heterophyllus* | - | 3.05 ± 0.11 |
| *Osmanthus* | *heterophyllus* | Goshiki | 3.05 ± 0.10 |
| *Osmanthus* | *heterophyllus* | Hariyama | 3.21 ± 0.09 |
| *Osmanthus* | *heterophyllus* | Head-Lee Fastigate | 3.12 ± 0.05 |
| *Osmanthus* | *heterophyllus* | Illicifolius | 3.12 ± 0.05 |
| *Osmanthus* | *heterophyllus* | Kaori Hime | 3.11 ± 0.12 |
| *Osmanthus* | *heterophyllus* | Kembu | 3.16 ± 0.12 |
| *Osmanthus* | *heterophyllus* | Ogon | 3.06 ± 0.09 |
| *Osmanthus* | *heterophyllus* | Purpureus | 3.06 ± 0.10 |
| *Osmanthus* | *heterophyllus* | Rotundifolius | 3.12 ± 0.03 |
| *Osmanthus* | *heterophyllus* | Sasaba | 2.99 ± 0.03 |
| *Osmanthus* | *heterophyllus* | Shien | 3.15 ± 0.04 |
| *Osmanthus* | *heterophyllus* | Variegatus | 3.05 ± 0.06 |

**Table 3.** Genome size of four *Osmanthus* species. Means followed by the same letter were not significantly different at the $\alpha$ = 0.05 significance level.

| Species | N | Genome size (pg) (Mean ± Standard Deviation) | Tukey Group |
|---|---|---|---|
| *O. armatus* | 3 | 3.07 ± 0.03 | ab |
| *O. fortunei* | 4 | 3.01 ± 0.04 | ab |
| *O. fragrans* | 7 | 2.94 ± 0.04 | b |
| *O. heterophyllus* | 13 | 3.10 ± 0.06 | a |

### 3.2. Floral Morphology

Thirty-three genotypes representing four species were observed for flowering and floral morphology between November 2017 and May 2018. Twenty-seven genotypes flowered during this period (Table 4). All observed *O. armatus* genotypes had perfect flowers and all observed *O. fortunei* genotypes had male-only flowers. Of 13 flowering *O. heterophyllus* genotypes, eleven were perfect while two ('Kaori Hime' and 'Ogon') were male-only (Figure 1). All flowering *O. fragrans* were male-only, but two genotypes, 'Apricot Echo' and an unnamed seedling, produced at least two flower types: male-only and functionally male with evidence of a reduced pistil (Figure 2). All male-only flowers observed in this study possessed reduced pistils were leafy and rounded in appearance and lacking a stigma. The perfect and male-only flowers of *O. armatus*, *O. fortunei*, and *O. heterophyllus* produced dehiscent pollen while all *O. fragrans* had anthers that were

fused with the corolla and lacked dehiscent pollen. Some pollen adhering to anthers was found in five of the seven *O. fragrans*, while the other two lacked pollen completely (Table 4).

**Table 4.** Floral morphology and presence or absence of pollen for four species of *Osmanthus* in McMinnville, TN, USA. A dash (-) indicates an unnamed seedling. Missing values indicate the accession did not flower during the observation period.

| Genus | Species | Cultivar | Flowers: Male (M) or Perfect (P) | Produced Pollen: Y/N | Female Structures: Normal or Reduced |
|---|---|---|---|---|---|
| *Osmanthus* | *armatus* | - | | | |
| *Osmanthus* | *armatus* | Jim Porter | P | Y | Normal |
| *Osmanthus* | *armatus* | Longwood | P | Y | Normal |
| *Osmanthus* | *fortunei* | - | M | Y | Reduced/leafy |
| *Osmanthus* | *fortunei* | - | | | |
| *Osmanthus* | *fortunei* | Fruitlandi | M | Y | Reduced/leafy |
| *Osmanthus* | *fortunei* | San Jose | M | Y | Reduced/leafy |
| *Osmanthus* | *fragrans* | - | M | Y | Reduced/leafy |
| *Osmanthus* | *fragrans* | - | | | |
| *Osmanthus* | *fragrans* | - | M or P | Y | Normal, reduced/leafy, and combination |
| *Osmanthus* | *fragrans* | Apricot Echo | M or P | Y | Normal, reduced/leafy, and combination |
| *Osmanthus* | *fragrans* | Beni Kin Mokusei | M | Y | Reduced/leafy |
| *Osmanthus* | *fragrans* | Thunbergii Clemson Hardy | | | |
| *Osmanthus* | *fragrans* | Fudingzhu | M | Y | Reduced/leafy |
| *Osmanthus* | *fragrans* | Quinnan Guifei | | | |
| *Osmanthus* | *fragrans* | Thunbergii | M | Y | Reduced/leafy |
| *Osmanthus* | *fragrans* | Tianxiang Taige | M | N | Reduced/leafy |
| *Osmanthus* | *fragrans* | Xiaoye Fondingzhu | M | | |
| *Osmanthus* | *fragrans* | Yinbi Shuanghui | M | N | Reduced/leafy |
| *Osmanthus* | *heterophyllus* | - | P | Y | Normal |
| *Osmanthus* | *heterophyllus* | Goshiki | | | |
| *Osmanthus* | *heterophyllus* | Hariyama | P | Y | Normal |
| *Osmanthus* | *heterophyllus* | Head-Lee Fastigate | P | Y | Normal |
| *Osmanthus* | *heterophyllus* | Illicifolius | P | Y | Normal |
| *Osmanthus* | *heterophyllus* | Kaori Hime | M | Y | Reduced/leafy |
| *Osmanthus* | *heterophyllus* | Kembu | P | Y | Normal |
| *Osmanthus* | *heterophyllus* | Ogon | M | Y | Reduced/leafy |
| *Osmanthus* | *heterophyllus* | Purpureus | P | Y | Normal |
| *Osmanthus* | *heterophyllus* | Rotundifolius | P | Y | Normal |
| *Osmanthus* | *heterophyllus* | Rotundifolius Nouvea | P | Y | Normal |
| *Osmanthus* | *heterophyllus* | Sasaba | P | Y | Normal |
| *Osmanthus* | *heterophyllus* | Shien | P | Y | Normal |
| *Osmanthus* | *heterophyllus* | Variegatus | P | Y | Normal |

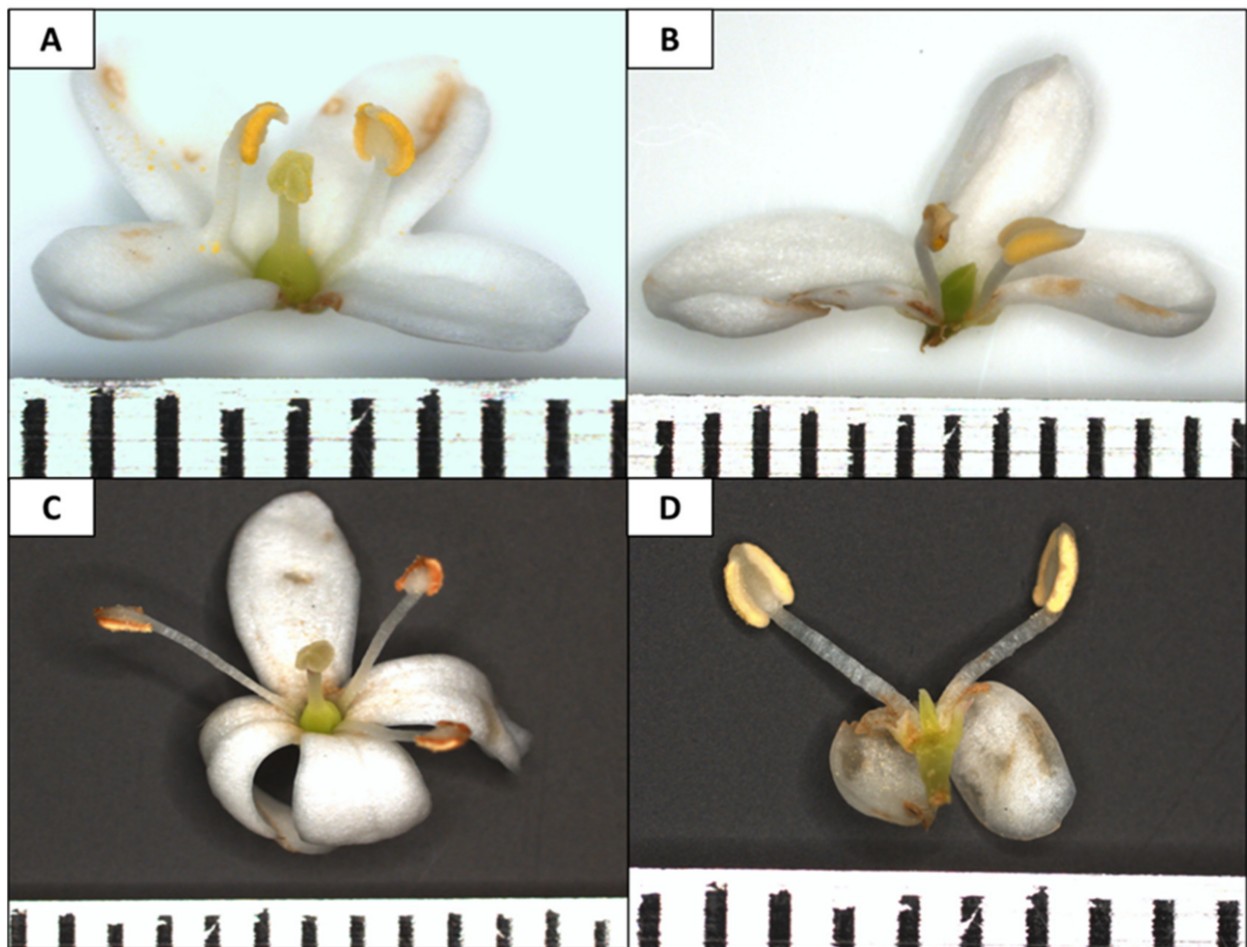

**Figure 1.** Flower of *Osmanthus armatus* 'Longwood'—perfect flower (**A**), *O. fortunei* unnamed seedling—male-only flower (**B**), *O. heterophyllus* 'Purpureus'—perfect flower (**C**), and *O. heterophyllus* 'Kaori Hime'—male-only flower (**D**). Petals were removed from (**A**,**B**,**D**) to fully expose pistils (**A**,**C**) or pistillodes (**B**,**D**). Flowers were photographed between 11 November 2017 and 9 January 2018 in McMinnville, TN, USA under a light microscope at 2× magnification with an Olympus Q Color 5 digital camera.

*3.3. Pollen Viability*

Twelve genotypes representing four species were used to examine pollen viability via staining. Flowers were collected between 13 November 2017 and 9 January 2018. Pollen viability was uniformly high, with 9 of 12 genotypes showing at least 80% viable pollen based on staining (Table 5). Cultivars of *O. heterophyllus* produced significantly less viable pollen than other species. Most notably, 'Shien', 'Ogon', and 'Variegatus'—all variegated *O. heterophyllus* cultivars—produced the least amount of viable pollen of the observed cultivars. Pollen did not dehisce from *O. fragrans* cultivars Apricot Echo and Fudingzhu and had to be removed with a scalpel. The mean number of pollen grains per field of view was lower for these cultivars (27 ± 8.6) than for cultivars of other species (63 ± 45; data not shown), but viability was 96% and 93% for Apricot Echo and Fudingzhu, respectively (Figure 3). Of the twelve observed genotypes, six produced male-only flowers and six produced perfect flowers; there was no difference in pollen viability between these flower types.

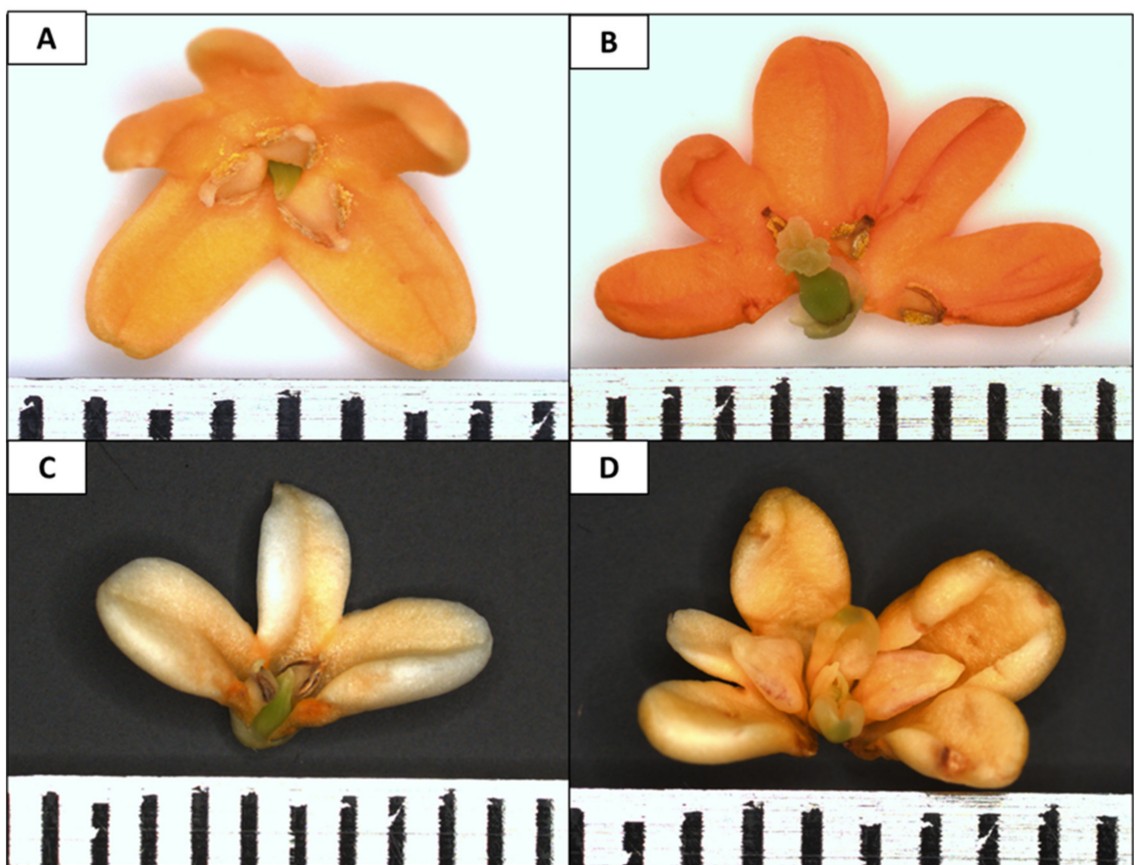

**Figure 2.** Flower of *Osmanthus fragrans* 'Apricot Echo' (**A**—male-only flower, **B**—perfect flower), 'Fudingzhu' male-only flower (**C**), and 'Tianxiang Tiage'—male-only flower (**D**). Petals were removed from B and C to fully expose pistils (**B**) or pistillodes (**A,C,D**). Flowers were photographed between 11 November 2017 and 9 January 2018 in McMinnville, TN, USA under a light microscope at 2× magnification with an Olympus Q Color 5 digital camera.

**Table 5.** Pollen viability of *Osmanthus* species and cultivars. Means followed by the same letter were not significantly different at the α = 0.05 significance level.

| Species | Stained Pollen (%) (Mean ± Standard Deviation) | | Germinated Pollen (%) (Mean ± Standard Deviation) | |
|---|---|---|---|---|
| *O. armatus* | 93.7 ± 8.10 | a | 44.0 ± 17.0 | a |
| *O. fortunei* | 88.9 ± 12.1 | a | 10.3 ± 4.00 | b |
| *O. fragrans* | 93.8 ± 5.50 | a | 42.3 ± 9.83 | a |
| *O. heterophyllus* | 71.3 ± 26.9 | b | 17.0 ± 17.8 | b |
| Cultivars | | | | |
| *O. fortunei* unnamed seedling | 97.9 ± 0.76 | a | 8.92 ± 1.01 | c |
| *O. fragrans* 'Apricot Echo' | 96.2 ± 6.42 | a | 43.9 ± 7.53 | a |
| *O. armatus* 'Jim Porter' | 95.8 ± 3.69 | a | 52.9 ± 8.67 | a |
| *O. fragrans* 'Fudingzhu' | 93.1 ± 5.20 | ab | 40.6 ± 13.3 | a |
| *O. armatus* 'Longwood' | 92.0 ± 10.3 | ab | 35.1 ± 20.3 | ab |
| *O. fortunei* 'Fruitlandii' | 86.3 ± 12.6 | ab | 12.1 ± 6.15 | bc |
| *O. heterophyllus* 'Rotundifolius' | 81.4 ± 13.9 | ab | 42.0 ± 5.96 | a |
| *O. heterophyllus* 'Kaori Hime' | 81.0 ± 21.2 | ab | 6.19 ± 3.47 | c |
| *O. heterophyllus* 'Hariyama' | 80.1 ± 1.60 | ab | - | |
| *O. heterophyllus* 'Shien' | 76.7 ± 10.8 | ab | 27.7 ± 6.04 | ab |
| *O. heterophyllus* 'Ogon' | 71.8 ± 22.3 | b | 47.7 ± 0.82 | a |
| *O. heterophyllus* 'Variegatus' | 8.00 ± 3.20 | c | - | |

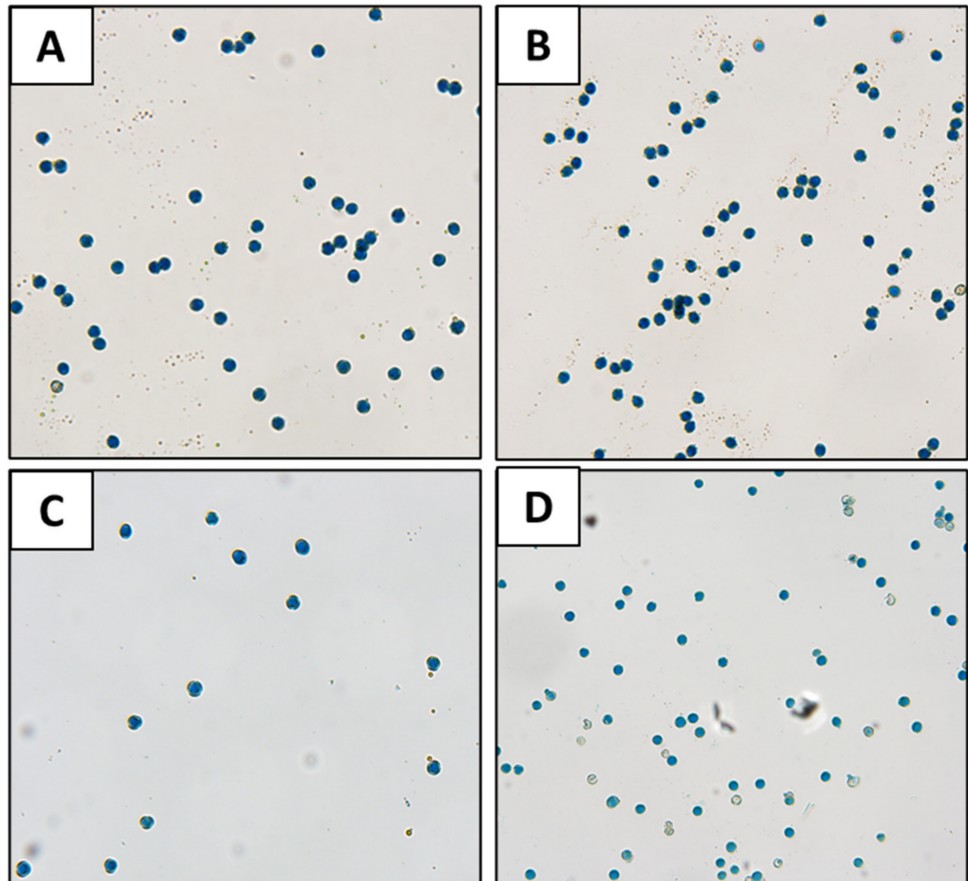

**Figure 3.** Stained pollen of *Osmanthus armatus* 'Jim Porter' (**A**), *O. fortunei* (**B**), *O. fragrans* 'Fudingzhu' (**C**), and *O. heterophyllus* 'Kaori Hime' (**D**). Fresh pollen was stained with modified Alexander's stain and observed after 30 min at 10× magnification using an Olympus BX-60 compound microscope with an Olympus Q Color 5 digital camera for image capture.

Ten cultivars were used for further pollen viability analysis using in vitro germination on PGM. Rates of pollen germination were about one-half to one-third rates of pollen viability estimated through staining (Table 5). Species and cultivars were ranked similarly except for *O. fortunei* where both cultivars showed less than 15% germination on PGM despite high rates of viability estimated through staining. There was no difference in germination rate between flower types (male-only vs. perfect).

### 3.4. Pollen Tube Growth

Pollen germinated on the stigma of all flowers in this study, including self-pollinations (Figures 4 and 5). There was a significant association between cross type and percentage of flowers with pollen tubes reaching the ovaries ($\chi^2 = 26.5$, $p < 0.001$). After 72 h, pollen tubes had reached the ovaries of in 67% of intraspecific crosses, 78% of interspecific crosses, and 0% of self-crosses. Cross type had a significant influence on pollen tube length 24 h (F = 16.5, $p < 0.0001$), 48 h (F = 14.6, $p < 0.0001$), and 72 h (F = 25.8, $p < 0.0001$) after pollination (Table 6). At all time points post-pollination, interspecific crosses produced the longest pollen tubes, followed by intraspecific crosses, and self-pollinations (Table 7). By 72 h post-pollination, mean pollen tube lengths were 1.6 ± 0.39 mm, 0.99 ± 0.95 mm, and 0.067 ± 0.070 mm for interspecific, intraspecific, and self-crosses, respectively. These comparisons were made adjusting for flower length, which accounted for a significant amount of variation in pollen tube length at 48 h post-pollination only (F = 6.6, $p = 0.014$) (Table 6).

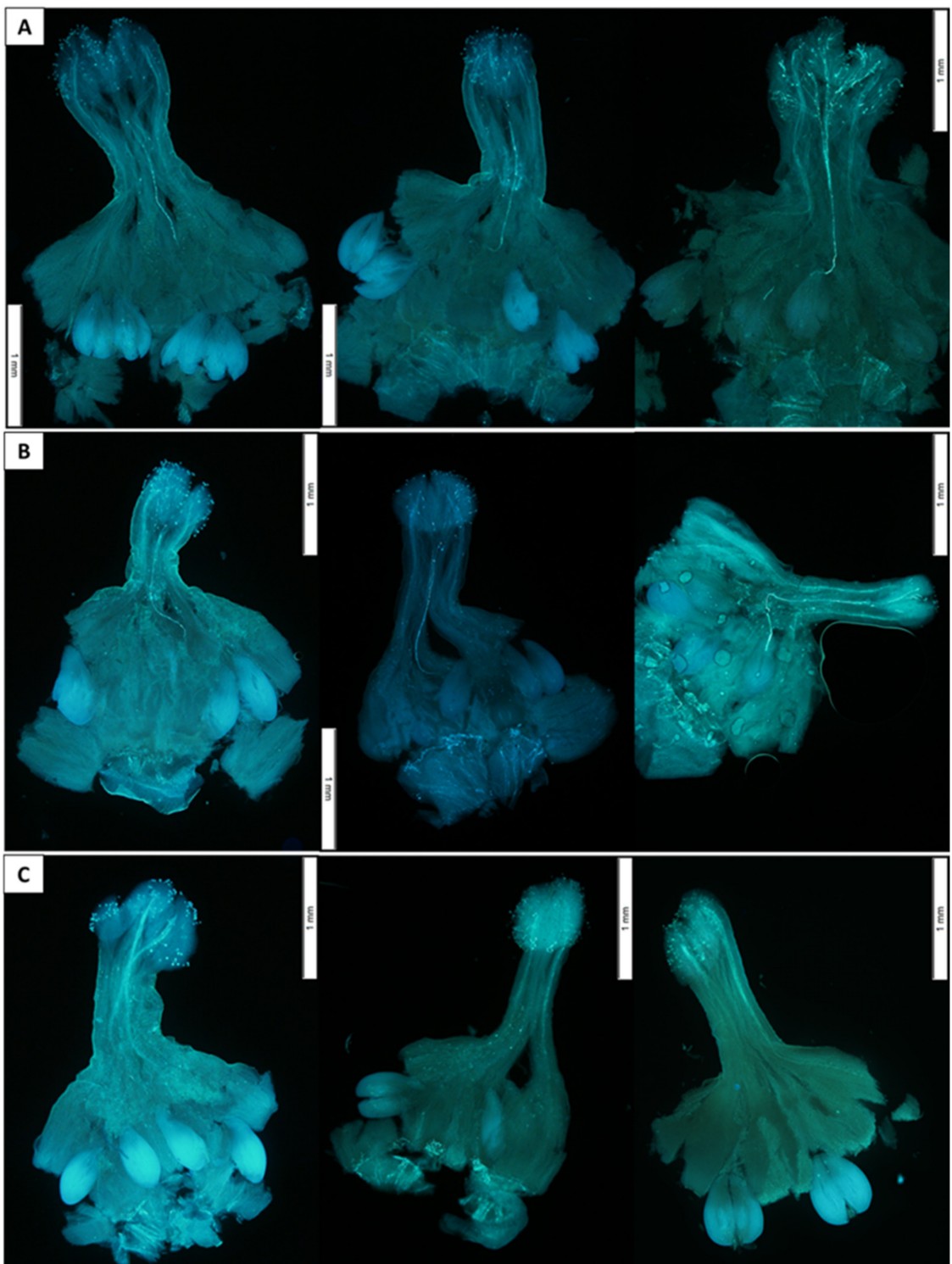

**Figure 4.** Stained pistils showing pollen tube growth from (**A**) intraspecific, (**B**) interspecific, and (**C**) self crosses of *Osmanthus armatus* 24 h (left), 48 h (middle), and 72 h (right) post-pollination. Crosses shown from top to bottom are: *O. armatus* 'Jim Porter' × *O. armatus* 'Longwood', *O. armatus* 'Jim Porter' × *O. heterophyllus* 'Kaori Hime', and *O. armatus* 'Jim Porter'. Pistils were fixed, rinsed, hydrolyzed, stained with decolorized aniline blue, and placed on microscope slides. Slides were observed after 1 h using an Olympus BX-60 compound microscope with an Olympus Q Color 5 digital camera for image capture.

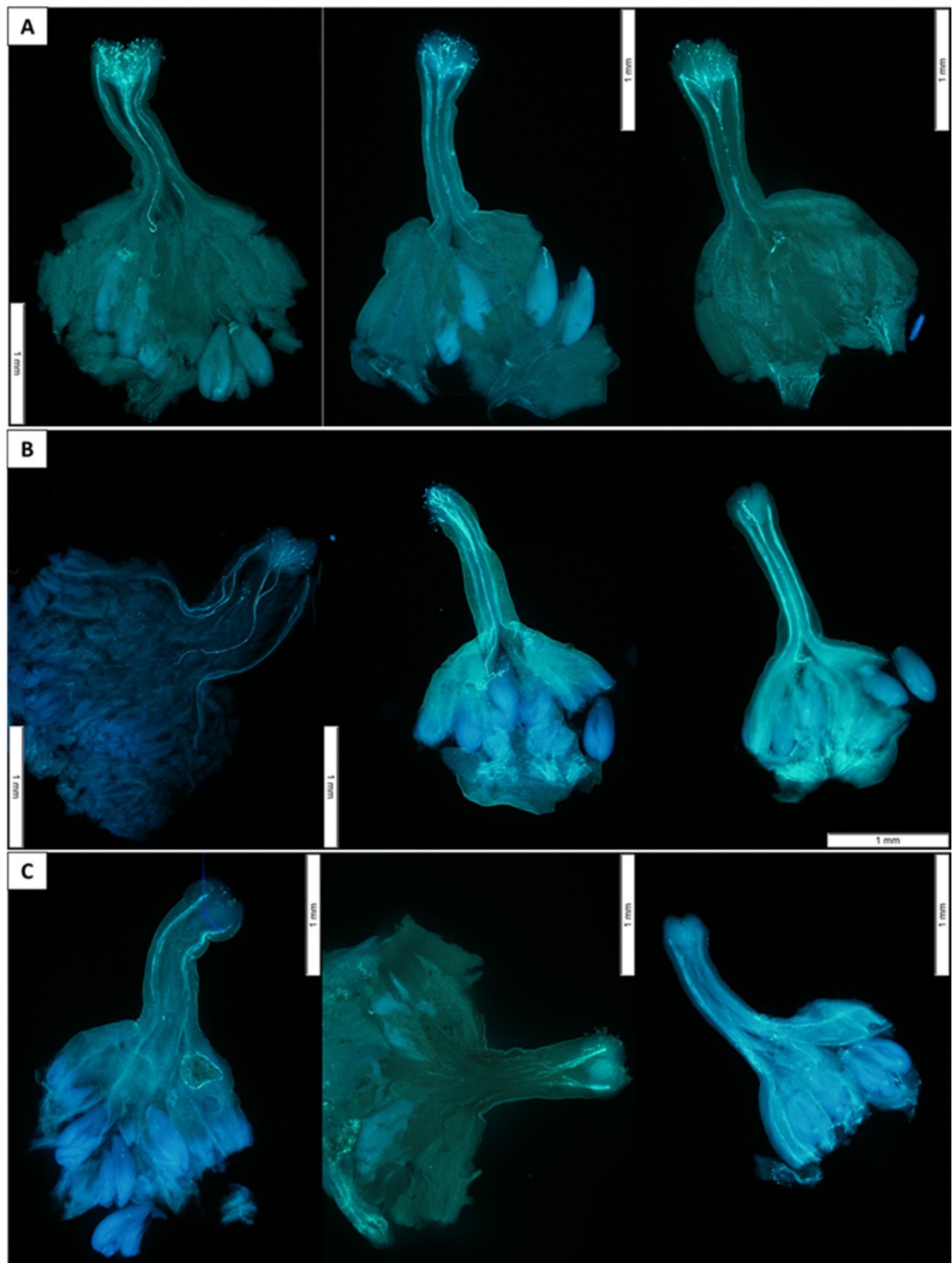

**Figure 5.** Stained pistils showing pollen tube growth from (**A**) intraspecific, (**B**) interspecific, and (**C**) self crosses of *Osmanthus heterophyllus* 24 h (left), 48 h (middle), and 72 h (right) post-pollination. Crosses shown from top to bottom are: *O. heterophyllus* 'Rotundifolius' × *O. heterophyllus* 'Kaori Hime', *O. heterophyllus* 'Rotundifolius' × *O. armatus* 'Longwood', and *O. heterophyllus* 'Rotundifolius'. Pistils were fixed, rinsed, hydrolyzed, stained with decolorized aniline blue, and placed on microscope slides. Slides were observed after 1 hour using an Olympus BX-60 compound microscope with an Olympus Q Color 5 digital camera for image capture.

**Table 6.** Analysis of covariance for sources of variation in pollen tube length for cross type (intraspecific, interspecific, or self) between *Osmanthus* species.

| Time (h) | Source | DF | Mean Square | F Value | Pr > F |
|---|---|---|---|---|---|
| 24 | Cross Type | 2 | 4.56 | 16.52 | <0.0001 |
| | Flower Length | 1 | 0.69 | 2.5 | 0.12 |
| | Error | 43 | 0.28 | | |
| 48 | Cross Type | 2 | 5.69 | 14.6 | <0.0001 |
| | Flower Length | 1 | 2.57 | 6.58 | 0.014 |
| | Error | 45 | 0.39 | | |
| 72 | Cross Type | 2 | 9.29 | 25.79 | <0.0001 |
| | Flower Length | 1 | 0.10 | 0.28 | 0.59 |
| | Error | 42 | 0.36 | | |

The ovaries of *O. armatus* and *O. heterophyllus* are divided into four ovules. Only one pollen tube reached the ovary in a flower, no matter how many pollen grains germinated on the stigmatic surface (Figures 4 and 5). Thus, only one ovule per flower is pollinated, resulting in a single seed per flower. The absence of self-pollen tubes in the ovaries indicates a self-incompatibility system in *Osmanthus*. Because pollen tubes grew slightly after self-pollination (Table 7), it appears that incompatible pollen tubes were inhibited after germination inside the stigma tissue rather than on the surface of the stigma.

**Table 7.** Average pollen tube length (mm) and percentage of flowers with pollen tubes reaching the ovary (in italics) for three cross types between *Osmanthus* species collected 24, 48, or 72 h post-pollination. Abbreviations: A = *Osmanthus armatus*, F = *O. fragrans*, H = *O. heterophyllus*, s = self. The female parent in a cross is listed first. Means followed by the same letter were not significantly different at the α = 0.05 significance level.

| | | | Hours after Pollination | | | | | | | | |
|---|---|---|---|---|---|---|---|---|---|---|---|
| | | | 24 | | | 48 | | | 72 | | |
| Cross Type | Cross Name | Parent Cultivars | Pollen Tube Length (mm) (Mean ± Standard Deviation) | | Flowers with Pollen Tubes in Ovary (%) | Pollen Tube Length (mm) (Mean ± Standard Deviation) | | Flowers with Pollen Tubes in Ovary (%) | Pollen Tube Length (mm) (Mean ± Standard Deviation) | | Flowers with Pollen Tubes in Ovary (%) |
| Interspecific | AxF | Jim Porter × Fudingzhu | 1.69 ± 0.338 | a | 33.3 | 1.31 ± 1.06 | ab | 33.3 | 1.96 ± 0.469 | a | 66.7 |
| Interspecific | AxH | Longwood × Kaori Hime | 1.33 ± 0.329 | a | 0 | 1.63 ± 0.418 | a | 66.7 | 1.74 ± 0.509 | a | 66.7 |
| Interspecific | HxA | Rotundifolius × Longwood | 0.851 ± 0.789 | a | 0 | 1.14 ± 0.747 | ab | 60 | 1.76 ± 0.328 | a | 100 |
| Interspecific | HxF | Shien × Fudingzhu | 1.12 ± 0.28 | a | 0 | 1.09 ± 0.772 | ab | 33.3 | 1.57 ± 0.176 | a | 66.7 |
| Intraspecific | AxA | Jim Porter × Longwood | 0.623 ± 0.596 | a | 14.3 | 0.999 ± 0.672 | ab | 33.3 | 1.19 ± 1.12 | ab | 50 |
| Intraspecific | AxA | Longwood × Jim Porter | 0 ± 0.0264 | a | 0 | 0 ± 0.098 | b | 0 | 0 ± 0.00577 | c | 0 |
| Intraspecific | HxH | Rotundifolius × Kaori Hime | 1.28 ± 0.46 | a | 16.7 | 1.42 ± 0.834 | a | 50 | 1.88 ± 0.103 | a | 100 |
| Intraspecific | HxH | Shien × Kaori Hime | 0.996 ± 0.795 | a | 0 | 0.959 ± 0.565 | ab | 33.3 | 0 ± 0 | c | 66.7 |
| Self | Axs | Jim Porter × self | 0.071 ± 0.0586 | a | 0 | 0.256 ± 0.01 | ab | 0 | 0.379 ± 0.0757 | bc | 0 |
| Self | Axs | Longwood × self | 0 ± 0.0479 | a | 0 | 0 ± 0.065 | b | 0 | 0 ± 0.0359 | c | 0 |
| Self | Hxs | Rotundifolius × self | 0 ± 0 | a | 0 | 0 ± 0.044 | b | 0 | 0.026 ± 0.049 | c | 0 |
| Self | Hxs | Shien × self | 0.21 ± 0.046 | a | 0 | 0.181 ± 0 | ab | 0 | 0.364 ± 0.0693 | bc | 0 |

## 4. Discussion

Genome size, floral morphology, and SI systems are properties of plants that may serve as barriers to hybridization [22,23]. All *Osmanthus* accessions plants investigated herein were diploid, indicating that no ploidy barriers should prevent mating among these *Osmanthus* species. The germination of the pollen on the stigma of self-pollinations with subsequent termination of pollen tube growth in the style provides support for a sporophytic SI system in *Osmanthus* [9]. Cultivated olive trees, a well-studied *Osmanthus* relative, also have a sporophytic SI system. This system is controlled by a single gene with two alleles that is responsible for all SI phenotypes observed in olive trees [13]. As related species with SI often share SI systems [7,9], it is likely that the sporophytic SI system observed in *Osmanthus* is also under single gene control. More crosses (including reciprocal crosses) are necessary to elucidate the number of alleles present in the *Osmanthus* SI system.

A major challenge to *Osmanthus* breeding improvement in the United States is the lack of female or perfect flowers in commercially available material, and cultivars of *O. armatus*, *O. fortunei*, and *O. heterophyllus*, along with genotypes of *O. fragrans*, were used herein to investigate mechanisms of pollination and incompatibility essential to hybrid breeding for these species. Perfect flowers were produced by both *O. armatus* cultivars that flowered and eleven out of thirteen *O. heterophyllus* cultivars. None of the *O. fragrans* or *O. fortunei* cultivars observed herein produced functional female flowers. As *O. fortunei* is an *O. heterophyllus* × *O. fragrans* hybrid, it is possible that male-only flowers may be a dominant trait, or simply that genotypes with male flowers are more attractive. If directional crosses only use *O. fragrans* as a male, maternal effects are difficult to understand. For example, none of the *O. fortunei* genotypes observed herein retained the smooth leaf form of its *O. fragrans* male parent. Hybrid populations using *O. fragrans* as both female and male parents are necessary to fully understand how important traits are inherited in *Osmanthus*.

*Osmanthus fragrans* 'Apricot Echo' and an unnamed seedling produced flowers with apparently functional pistils, but resultant seed that appeared viable has failed to germinate (personal observation). Olive trees have similar floral morphology where cultivars have either male-only or perfect flowers. The ratio of male-only to perfect flowers is controlled genetically and environmentally; and varies across years, trees, and even among shoots and inflorescences of the same tree [12]. In this study, three clones of 'Apricot Echo' were observed with similar flower types, but the influence of environment cannot be tested as these were grown in a single location. Opportunity to introduce more *O. fragrans* genotypes into the U.S. market is promising, as over 120 cultivars (including those that produce perfect flowers) have been described [2].

The *Osmanthus* cultivars in this study produced viable pollen, had pollen tubes reach the ovaries in all cross combinations attempted, and do not need to be emasculated due to strong self-incompatibility. However, each flower produces only a single seed, many ideal cross combinations are limited by the availability of female flowers, and germination may be uneven, further limiting the evaluation of cross success [24]. *Osmanthus fragrans* is further divided into four groups based on flowering time: Albus, Luteus, and Aurantiacus Groups flower in Autumn while the Asiaticus Group flowers year-round excepting summer. Cultivars with perfect flowers and with male only flowers exist in all four groups [2,25]. Extending flowering time by photoperiod and/or cultural manipulation will be necessary for producing diverse breeding lines, as populations and cultivars of all four groups are valuable as sources of genetic variation in flower color and flowering time.

Fragrance is a valuable trait in ornamental landscape trees, and few flowering tree species boast the level of fragrance found in *Osmanthus* species, especially *O. fragrans* [1]. These species are used prolifically as specimen trees, hedges, and fragrant verges in areas where they thrive [1]. Further breeding improvement is necessary in *Osmanthus* to increase cold-hardiness and introgress the smooth leaf and upright form of *O. fragrans* into cultivars adapted for U.S. landscapes. Current breeding efforts in the U.S. should focus on importing or developing cultivars and populations with perfect flowers, lengthening flowering times

through photoperiod and/or temperature, and improving germination rates of seed. These traditional plant breeding methods appear to be the bottleneck in a breeding system where there is good compatibility among species and cultivars. Producing new fortune's osmanthus hybrids as well as hybrids of fragrant tea olive (*O. fragrans*) and sweet olive (*O. armatus*) will provide variation for selecting new cultivars across a wide range of environments. Tools to select parents and verify hybrids, including molecular markers and a transcriptome data set for identification of important genes, are now available for fragrant tea olive and its relatives [25–27]. Novel hybrids allied with these advanced breeding tools will be foundational for the expanded use of *Osmanthus* into new or colder areas.

**Funding:** This research and APC was funded by the Floral and Nursery Research Institute and USDA-ARS in-house project 8020-21000-072-00D.

**Data Availability Statement:** The data presented in this study are available on request from the corresponding author.

**Acknowledgments:** The author would like to thank Carrie Witcher for assistance with pollen collection and microscopy. Benjamin Moore and Joseph Davis maintained the plants used in this experiment. Mention of trade names of commercial products in the publication is solely for the purpose of providing specific information and does not imply recommendation or endorsement by the U.S. Department of Agriculture. The raw data supporting the conclusions of this manuscript will be made available by the author, without undue reservation, to any qualified researcher.

**Conflicts of Interest:** The author declares no conflict of interest.

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
