# Peer review of "Genome Size, Flowering, and Breeding Compatibility in Osmanthus Accessions"

_horticulturae, doi:10.3390/horticulturae9010056_

Round 1
Reviewer 1 Report
Comments and suggestions for authors
Title: Title should be corrected as "Genome size, flowering, and breeding compatibility in Osmanthus accessions"
Abstract: When and where did the evaluation (experiment) carried out?
Introduction: It is well-written
Materials and Methods:What is the experimental design and how many replications were used in the experiment?
Results: Provide CV(%) and P value in table 2.
Provide CV(%) in table 3.
Provide CV(%) in table 5.
It would be better to write ' Pollen tube growth' as sub-headings rather than growth only.
Provide CV(%) in table 7.
Write column heading with unit of measurement for the italic values of the column in table 7.
Discussion: It is well-written
References: All references should follow the journal guidelines.
Provide DOI for journal references.

Author Response
Thank you for your comments. Please find my responses in bold, along with a clean copy of the manuscript. A copy of the manuscript with changes tracked is available at your request.
- Title: Title should be corrected as "Genome size, flowering, and breeding compatibility in Osmanthus accessions"
Changed as suggested.
- Abstract: When and where did the evaluation (experiment) carried out?
Added “…in McMinnville, TN, USA…” where indicated.
- Materials and Methods:What is the experimental design and how many replications were used in the experiment?
To “Pollen Staining” the following was added, “Pollen was sampled three times: 13 – 14 November 2017, 07 – 08 December 2017, and 09 January 2018 for a total of 9 flowers per genotype (three flowers at three time points). Genotypes were arranged in a completely randomized design.” To “pollen in vitro germination”, the following was added, “Three replicates (dishes) per accession were used in tests of pollen germination. Replicates were arranged in a completely randomized design.”
- Results: Provide CV(%) and P value in table 2.
For all genome size estimates, the following was added to the materials and methods: “For each sample, at least 3000 nuclei were analyzed revealing a single peak with a coefficient of variation (CV) less than 4.9%.”. For means in general, standard deviation is presented as a measure of between-measurement variation for each group, as is standard for this journal.”
- Provide CV(%) in table 3.
For all genome size estimates, the following was added to the materials and methods: “For each sample, at least 3000 nuclei were analyzed revealing a single peak with a coefficient of variation (CV) less than 4.9%.”. For means in general, standard deviation is presented as a measure of between-measurement variation for each group, as is standard for this journal.”
- Provide CV(%) in table 5.
Standard deviation is presented as a measure of between-measurement variation for each group, as is standard for this journal. Please advise if a different CV is being requested here.
- It would be better to write ' Pollen tube growth' as sub-headings rather than growth only.
Changed as suggested.
- Provide CV(%) in table 7.
Standard deviation is presented as a measure of between-measurement variation for each group, as is standard for this journal. Please advise if a different CV is being requested here.
- Write column heading with unit of measurement for the italic values of the column in table 7.
Changed as suggested.
- References: All references should follow the journal guidelines.
All references were formatted for journal guidelines.
- Provide DOI for journal references.
DOIs were included as suggested.

Reviewer 2 Report
The manuscript "Genome size, flowering, and breeding compatibility in Osmanthus" measured genome size, pollen viability and in vitro pollen germination, and self-compatibility across Osmanthus species. The experiments are well-designed and performed, and the conclusions are appropriate to the results presented. This study provides insights into opportunities and challenges for Osmanthus hybrid breeding. I don’t have any main concerns about the manuscript, so I think is suitable to be published before some minor edits were made.
Please add a space before “pollinated flowers” in the last sentence of the second paragraph of “Controlled pollinations”.
Figure 3, it looks like the pollen size of panel D is significantly smaller than the others. Were all pictures taken under the scale? If yes, is there a correlation between genome and pollen size? Some investigators claimed that doubling the diploid chromosome number increases pollen grain size in maize.
Make sure the first letter of “growth” on page 11 is capitalized.
Table 7, under the “Parent cultivars” column, it might be good to indicate which one is female or male in the crosses.
Author Response
Thank you for your comments. Please find my responses in bold, along with a clean copy of the manuscript. A copy of the manuscript with changes tracked is available at your request.
- The manuscript "Genome size, flowering, and breeding compatibility in Osmanthus" measured genome size, pollen viability and in vitro pollen germination, and self-compatibility across Osmanthus species. The experiments are well-designed and performed, and the conclusions are appropriate to the results presented. This study provides insights into opportunities and challenges for Osmanthus hybrid breeding. I don’t have any main concerns about the manuscript, so I think is suitable to be published before some minor edits were made.
Thank you for your comments. They improved the manuscript considerably.
- Please add a space before “pollinated flowers” in the last sentence of the second paragraph of “Controlled pollinations”.
Changed as suggested.
- Figure 3, it looks like the pollen size of panel D is significantly smaller than the others. Were all pictures taken under the scale? If yes, is there a correlation between genome and pollen size? Some investigators claimed that doubling the diploid chromosome number increases pollen grain size in maize.
Unfortunately we did not measure pollen size. All of these accessions are diploid. It is worth noting that the cultivar in panel D (which looked like the smallest pollen) has the largest genome of those pictured. It is likely that there is natural variation among species, and cultivars within a species, for pollen size.
- Make sure the first letter of “growth” on page 11 is capitalized.
Changed to “Pollen tube growth”.
- Table 7, under the “Parent cultivars” column, it might be good to indicate which one is female or male in the crosses.
Added to the title of Table 7, “The female parent in a cross is listed first.”

Reviewer 3 Report
Dear author,
Your study is well conceptualized but major revision is required. Please, add more general text about genome size, floral morphology, pollen viability, and self- and cross-compatibility in Introduction section. Also, you have to improve text in discussion section. I strongly suggest that you compare your results with more other similar studies and consequently add more other references at the manuscript. All comments are provided at the pdf of the manuscript.
I look forward to seeing your revised version of the manuscript.

Author Response
Thank you for your comments; they greatly improved the manuscript. Please find my responses in bold, along with a clean copy of the manuscript. A copy of the manuscript with changes tracked is available at your request.
Reviewer 3
Page: 1
- In the present study you investigated genome size in five species.
We investigated genome size in five species, but floral morphology, pollen viability, and compatibility in only four species. This is because one species, O. delavayi, was only represented by a cutting sent by the U.S. National Arboretum. Since O. delavayi is not mentioned again in this paper, we removed it from the genome size date to reduce confusion.
- In my opinion this sentence should be moved before the previous sentence " Pollen viability based on staining and in vitro germination ranged from 8% to 98% and 6% to 53%, respectively "
Changed as suggested.
- In this section, the general text about genome size, floral morphology and breeding compatibility is missed.
The introduction was rewritten, with the following text added, “The unique reproductive systems found across plant species play a major role in the techniques, timing, and strategies of plant breeding efforts [7]. Approximately 40% of angiosperms, including many woody trees and shrubs, have mechanisms which limit or prevent self-fertilization [8]. Self-incompatibility (SI) refers to a collection of diverse mechanisms used by plants to reject self-pollen or pollen from genetically related individuals to prevent inbreeding or promote outcrossing [8-10]. Self-incompatibility may result from a single multi-allelic gene or several genes [10]. In the two most common systems, SI is regulated by either the haploid genotype of the pollen grain (gametophytic incompatibility) or the diploid genotype of the pollen parent (sporophytic incompatibility) [8-10]. Because distinct individuals can share identical SI genotypes, incompatible crosses are not limited to self-pollination. SI systems can limit compatible matings among genotypes, limiting production potential where seed set is important and limiting successful crosses in breeding programs [7,11].
While the SI system of Osmanthus has not been documented, recent breeding studies in olive (Olea europaea) have uncovered a sporophytic SI system controlled by a single gene that regulates a series of cellular interactions at the stigmatic surface of the pistil [12]. In this system, SI is manifested by the inhibition of pollen hydration, pollen tube germination, or pollen tube infiltration of the stigmatic surface[7,12]. All olive cultivars examined are self-incompatible, though “leaky” SI has been reported in a few genotypes in the presence of large quantities of incompatible pollen [7]. Visualization of pollen germination and pollen tube growth allows for the differentiation between pre and post zygotic reproductive barriers which in turn influences the choice of plant breeding techniques (somatic hybridization or embryo culture) that may be used to mitigate incompatibility [13]…. Polyploidy – or whole genome duplication – can also serve as a reproductive barrier among otherwise compatible species, as gametes with uneven numbers of chromosome fail to unite at fertilization [16]. Almost 70% of flowering plants have undergone ploidy changes in their evolutionary history [17], such that ploidy complexes can be major barriers for breeding compatibility [16,18].”
- This paragraph is very similar with the first paragraph in publication Alexander, 2019.
This paragraph was removed and replaced with the current first two paragraphs.
- Line 29 You should provide reference about that.
Added Dirr, 2011.
- Line 30 – 31 You should provide reference about that.
Added Xiang and Liu, 2007
- At the beginning of the sentence you must use full species name.
Changed as suggested.
- Lines 35 – 39 You should provide reference about that.
Added Dirr, 2009.
- 7,8 and 9 in the United States (according to reference Alexander, 2019)
Changed to 7, 8, and 9
- Lines 42 – 44 Please, provide reference(s) about that.
Added Saumitou-Laprade et al., 2016
- This is information from master's thesis. Please, provide additional references from some scientific journals.
Added: Hu, 2019 and Yue et al., 2020
Page: 2
- Author: author Subject: Highlight Date: 11/28/2022 10:16:02 AM Please, check again the year.
The 6th edition came out in 2009. The edition number was added to the citation.
- Lines 50 – 51 Please, provide reference about that.
This sentence was removed when the Introduction was rewritten.
- Author: author reference(s)? Subject: Highlight Date: 11/28/2022 10:16:20 AM
Added Xu et al., 2014 for a more recent reference.
- "Thirty-four genotypes representing four species were evaluated for genome size, floral morphology, pollen viability, and self- and cross-compatibility"
Changed as suggested to say: “The objectives of this study were to evaluate genome size, floral morphology, pollen viability, and self- and cross-compatibility of Osmanthus.”
- genotypes
Added as suggested.
- Why the number and type of sources is not the same for all analayzed species?
We obtained all the Osmanthus that was freely or commercially available at the time this study started. Some species are more popular and therefore have more cultivars available. For example, there are many more cultivars of O. heterophyllus available on the market than O. armatus, and our numbers reflect that.
Page: 3
- In the Material and Methods section is wrote:.Twenty-six Osmanthus cultivars and eight unnamed accessions representing five species. You should precise how much cultivars and unnamed accessions were available for genom size determination. Also, why other 6 genotypes did not available for genom size determination?
Thank you for your comment. Specific numbers for each genotype were added as follows: “Twenty-eight genotypes representing five species were available for genome size determination; O. armatus (n = 3 genotypes), O. delvayi (n = 1 genotype), O. fortunei (n = 4) genotypes, O. fragrans (n = 12 genotypes, and O. heterophyllus (n = 14 genotypes).” The other six genotypes were accessions we received after genome sizing.
- Zea maize 5.67 Pisum sativum 9.09
Changed as suggested.
Page: 4
- Why did not you investigate genome size of O. fragrans (unnamed accessions, Daye Fondingzhu, Quinnan Guifei, Tianxiang Taige, Yinbi Shuanghui ), O. heterophyllus (Rotundifolius Nouvea)?
These cultivars are new to the American market, and were not available when genome size determinations were made.
- What about genome size of O. fortunei and O. armatus?
These were not significantly different than the genome size of O. fragrans.
- Author: author Subject: Highlight Date: 11/28/2022 10:17:12 AM How much this information is reliable because you have only one unnamed accession?
We have removed O. delavayi from this paper, as it was in the genome size section but nowhere else in the manuscript.
- At the Table 2 is written 3.09 ± 0.08.
We removed this value from the text and table.
- The number 0.06 is wrong. You should put there value 0.08 pg
The value shown in Table 3 (3.10 ± 0.06) is correctly written in the text. This the average value for the species, reported in Table 3.
- unnamed accession
We removed all references to the single unnamed accession of O. delavayi
- Inserted Text: A dash (-) represents an unnamed seedling.
Changed as suggested.
- Inserted Text: (mean ± standard deviation)
Changed as suggested.
Page: 6
- Author: author Subject: Inserted Text Date: 11/28/2022 11:27:21 AM (mean ± standard deviation)
Changed as suggested.
- Author: author Subject: Inserted Text Date: 11/28/2022 11:27:26 AM N
Changed as suggested.
- Please, put the significant p values, and for not significant p values you should put abbreviation ns (not significant).
Many recently published articles in Horticulturae use the Tukey groups in the tables, for example Horticulturae 2022, 8(11), 1094; https://doi.org/10.3390/horticulturae8111094, published 20 November 2022. If this format is not acceptable for this specific article, please provide more information on which p-values should be reported.
- Based on the table 2, there are 7 genotypes.
Thank you for the close review. Changed to 7 as suggested.
- In my opinion, you should write N in the table, because this is universal abbreviation for number.
Thank you for your comment. Changed as suggested.
- Why did not you investigate florel morphology in O. delavayi?
We received O. delavayi as a cutting; it died before flowering. To reduce confusion, we removed O. delavayi from this experiment (as it only had one accession).
- Based on information from Table 4, the information about flowers in Osmanthus armatus (-) was not reported.
This accession did not flower. Added to the table title: “Missing values indicate the accession did not flower during the observation period.”
- What about Osmanthus heterophyllus (Goshiki)?
This accession did not flower. Added to the table title: “Missing values indicate the accession did not flower during the observation period.”
- Based on information from Table 4, the information about flowers in Osmanthus fortunei (-) was not reported.
This accession did not flower. Added to the table title: “Missing values indicate the accession did not flower during the observation period.”
- In the Table 4 there are 14 observed O. heterophyllus genotypes.
That is correct, but one did not flower so there were 13 observed flowering.
- Author: author Subject: Inserted Text Date: 11/28/2022 11:28:31 AM eleven
Thank you. Changed to eleven.
- Author: author Subject: Inserted Text Date: 11/28/2022 11:28:39 AM Seven of twelve
To reduce confusion between the numbers of accessions observed and the number of accessions flowering, the word “observed” was changed to “flowering” in lines as follows: : Of 13 observed flowering O. heterophyllus genotypes, nine eleven were perfect while two (‘Kaori Hime’ and ‘Ogon’) were male-only (Figure 1). All flowering O. fragrans observed (n = 7) were male-only…”
- This is not reported in the Table 4.
Changed “NA78642” to “unnamed seedling”.
- What about O. fragrans (-), (Thunbergii Clemson Hardy), (Quinnan Guifei)?
These did not flower during the observation period. Added to the table title: “Missing values indicate the accession did not flower during the observation period.”
Page: 7
- -, Fruitlandi, San Jose???
Added “unnamed seedling” to figure caption.
- -perfect flower
Added as suggested.
- -male-only flower
Added as suggested.
- – 48: Format changes made as suggested.
- -perfect flower
Added as suggested.
- What about C?
No petals were removed from the flower in picture C.
- -male-only flower
Added as suggested.
Page: 8
- male-only or perfect flower
Added as suggested.
- male-only flower
Added as suggested.
- male-only flower
Added as suggested.
Page: 9
- Material and method section: Pollen was sampled three times: 13 – 14 November 2017, 07 – 08 December 2017, and 09 January 2018 for a total of 9 flowers per genotype (three flowers at three time
points).
Thank you for your comment. Changed “11 November” to “13 November”.
- 9 of 12
Changed as suggested.
- There are 6 cultivars of O. heterophyllus, and only per 2 cultivars of O. fragrans, O. fortunei and O. armatus. This is not balanced comparison.
There are many more genotypes of O. heterophyllus available than other Osmanthus species. Mixed models were used for analysis of variance specifically to mitigate the effects of an unbalanced design.
- Why did you think that this information was important to report?
From a plant breeder’s perspective, it is useful to know which cultivars may make good pollen parents and where barriers (like non-dehiscent pollen) might exist. It is also useful to those wishing to replicate the study – without scraping the anthers one wouldn’t get enough pollen to count.
- Please, check these information. In my opinion, which is based on data reported in Table 4, there are 5 genotypes that produced male-only flowers, six genotypes which produced perfect flowers and one genotype that produced male-only or perfect flowers.
This information on pollen viability is based on Table 5, where 12 genotypes were observed for pollen viability.
- In the table 5 there are twelve genotypes.
Thank you. Thirteen was changed to twelve.
Page: 10
- Why did not you use O. heterophyllus 'Hariyama' and O. heterophyllus 'Variegatus' for pollen viability analyses using in vitro germination on PGM?
There were not enough flowers for both analyses.
- (mean ± standard deviation) Author: author Subject: Inserted Text Date: 11/28/2022 11:30:41 AM (mean ± standard deviation)
Changed as suggested.
- Please, put the significant p values, and for not significant p values you should put abbreviation ns (not significant).
Many recently published articles in Horticulturae use the Tukey groups in the tables, for example Horticulturae 2022, 8(11), 1094; https://doi.org/10.3390/horticulturae8111094, published 20 November 2022. If this format is not acceptable for this specific article, please provide more information on which p-values should be reported.
Page: 11
- ??? I suppose that you would like to write: Pollen tube growth
Yes, thank you.
- Author: author Subject: Highlight Date: 11/28/2022 11:31:27 AM What about Osmanthus armatus? How did you calculate these percentage?
This paragraph gives means for cross types, and these means include both O. armathus and O. heterophyllus crosses. Referencing figure 5 made it seem like we were referring to O. heterophyllus, so we removed the reference to figure 5 from the text.
- Author: author Subject: Highlight Date: 11/28/2022 11:31:31 AM Please, check the time period which you put in the Table 6.
Changed to 2, 48, and 72. Thank you.
- Author: author Subject: Highlight Date: 11/28/2022 11:31:37 AM Please, add degrees of freedom.
Degrees of freedom are shown in Table 6.
- Author: author Subject: Highlight Date: 11/28/2022 11:31:41 AM F values are not the same in the text and the Table 6.
Thank you for your comment. F-values were changed to match the table.
- Author: author Subject: Highlight Date: 11/28/2022 11:31:45 AM Please, add degrees of freedom.
Degrees of freedom are shown in Table 6.
- Author: author Subject: Highlight Date: 11/28/2022 11:32:03 AM Please, add degrees of freedom.
Degrees of freedom are shown in Table 6.
- Author: author Subject: Highlight Date: 11/28/2022 11:32:18 AM Why did not you put these information in Table 7?
This information is in Table 7; it can be derived by averaging the means for each cross type.
- Author: author Subject: Highlight Date: 11/28/2022 11:32:22 AM ???
This should have been 72, and was changed accordingly.
- Author: author Subject: Highlight Date: 11/28/2022 11:32:30 AM You should add degrees of freedom.
Degrees of freedom are shown in Table 6.
- Author: author Subject: Inserted Text Date: 11/28/2022 11:32:37 AM (Table 6)
Added reference to Table 6 as suggested.
Page: 12
- Author: author Subject: Inserted Text Date: 11/28/2022 11:32:41 AM O. armatus ‘Jim Porter’
Changed as suggested.
Page: 13
- Author: author Subject: Inserted Text Date: 11/28/2022 11:32:45 AM O. heterophyllus ‘Rotundifolius’
Changed as suggested.
- Author: author Subject: Inserted Text Date: 11/28/2022 11:32:58 AM between
Changed as suggested.
Page: 14
- I suppose that this is the accidental mistake, because throughout the manuscript you mentioned 24, 48 and 72 hours but not 48, 72 and 96 hours.
Yes, the table and all text was corrected to say 24, 48, and 72 hours.
Page: 15
- Author: author Subject: Inserted Text Date: 11/28/2022 11:33:06 AM Abbreviations: A = Osmanthus armatus, F = O. fragrans, H = O. heterophyllus, s = self.
Moved to title as suggested.
- You should put significant p values and not significant p values can be labeled as ns.
Many recently published articles in Horticulturae use the Tukey groups in the tables, for example Horticulturae 2022, 8(11), 1094; https://doi.org/10.3390/horticulturae8111094, published 20 November 2022. If this format is not acceptable for this specific article, please provide more information on which p-values should be reported.
- All these information are reported in the Material and Method section.
Removed as suggested.
Page: 16
- Author: author Subject: Highlight Date: 11/28/2022 11:33:23 AM This sentences would be fit better in the Introduction section.
Many recently published articles in Horticulturae use the Tukey groups in the tables, for example Horticulturae 2022, 8(11), 1094; https://doi.org/10.3390/horticulturae8111094, published 20 November 2022. If this format is not acceptable for this specific article, please provide more information on which p-values should be reported.
- Author: author Subject: Highlight Date: 11/28/2022 11:33:33 AM Please, check the Table 4.
Changed to 11 out of 13.
- Author: author Subject: Highlight Date: 11/28/2022 11:33:36 AM What about O. fragrans -, and O. fragrans Apricot Echo?
These genotypes did not produce functional female flowers.
- This sentences would be better fir in the Introduction section.
Moved to Introduction as suggested.
- Please, explain that.
Added to explain maternal effects, “If directional crosses only use O. fragrans as a male, maternal effects are difficult to understand. For example, none of the O. fortunei genotypes observed herein retained the smooth leaf form of its O. fragrans male parent. Hybrid populations using O. fragrans as both female and male parents are necessary to fully understand how important traits are inherited in Osmanthus.”
- You should provide reference(s) for that statement.
Rewrote this statement to provide an empirical example rather than a statement, “For example, none of the O. fortunei genotypes observed here retained the smooth leaf form of its O. fragrans male parent.”
- This is not case only for O. fragrans "Apricot Echo", but also for O. fragrans "-".
Changed to say,” Osmanthus fragrans ‘Apricot Echo’ and an unnamed seedling produced flowers with apparently functional pistils, but resultant seed that appeared viable has failed to germinate (personal observation).”
- Whether this reference is actually: "Seifi E, Guerin J, Kaiser BN, and M. Sedgley. 2015. Flowering and fruit set in olive: a review. Iranian J Plant Physiol. 5(2):1263-1271" at reference list?
Yes. Changed to Seifi et al 2015 in text.
- You should give some explanation about that?
Changed “winter protection and other means” to say, “photoperiod and/or cultural manipulation” for clarity.
- What does that mean?
Changed “winter protection and other means” to say, “photoperiod and/or cultural manipulation” for clarity.
- Line 297 (reference(s))
Added Dirr, 2011.
- Line 298 (reference(s))
Added Dirr, 2011.
Page: 17
- Inserted Text Date: 11/28/2022 11:34:20 AM (O. fragrans)
Added as suggested.
- Inserted Text Date: 11/28/2022 11:34:24 AM (O. armatus)
Added as suggested.
- You should add more references which are relevant for the present research. Also, the order of the reference is not correlated with alphabetical order.
We added many new references regarding self-incompatibility systems in plants and olives specifically, cold-hardiness breeding in Osmanthus, and comparisons of our results to previous research. All references were ordered as per the journal style.
- You should try to write references based on alphabetical order. Also, this reference is wrong cited in the text.
All references were placed alphabetically and the Seifi reference was corrected in the text.
- Author: author 1990?
The 6th edition came out in 2009. The edition number was added to the citation.

Round 2
Reviewer 3 Report
Dear author,
The revised version of the manuscript is improved but I found that major revision is required. All comments are provide in the pdf of the manuscript.

Author Response
Response to review 2 - horticulturae-2082921
Thank you again for your review. Please find the author responses in bold, along with a tracked copy of the manuscript uploaded to the website. All format changes were incorporated. Please note that References are correctly numbered in the CLEAN copy only.
- Line 61: Approximately 40% of angiosperms - I could not find this information in article: Silva, N.F.; Goring, D.R. Mechanisms of self-incompatibility in flowering plants. Cell. Mol. Life Sci. 2001, 58, 1988-2007;
Thank you for your comment. The original source of the information is new Reference number 8, page 1 (Igic, B.; Lande, R.; Kohn, J.R. Loss of self-incompatibility and its evolutionary consequences. Intl. J. Plant Sci. 2008, 169, 93–104; DOI: 10.1086/523362.) and can also be found in Reference 7, page 1.
- Line 69-70: “Because distinct individuals can 69 share identical SI genotypes, incompatible crosses are not limited to self-pol-70 lination.” Please, rephrise this sentence because it is identical with sentence in reference 7 in your reference list.
Rephrased as suggested, to “Because individual plants can possess identical alleles at SI loci, SI systems can have consequences beyond self-pollination.”
- Line 81-84: “Visualization of pollen 81 germination and pollen tube growth allows for the differentiation between 82 pre and post zygotic reproductive barriers which in turn influences the choice 83 of plant breeding techniques (somatic hybridization or embryo culture) that 84 may be used to mitigate incompatibility.” Please, rephrase this sentense because it is the same as the sentense from Alexander, L. Ploidy level influences pollen tube growth and seed viability in interploidy crosses of Hydrangea macrophylla. Front. Plant Sci. 2020, 11, 100.
Rephrased as suggested to, “In order to understand the pre and post zygotic reproductive barriers at work in a species, pollen germination and pollen tube growth can be visualized using microscopy. The type of reproductive barriers in place influence the choice of plant breeding techniques that may be used to overcome SI.”
- Line 123: “Thirty-three accessions representing four species of Osmanthus used for genome sizing, floral morphology, 123 pollen viability, and/or incompatibility studies at the U. S. National Arboretum in McMinnville, Tennessee, USA. A 124 dash (-) indicates seedling (i.e., not a named cultivar). “ The sentence in abstract:"Thirty-four genotypes representing four species were evaluated in McMinnville, TN, USA for genome size, floral morphology, pollen viability, and self- and cross-compatibility. "
The abstract was changed to reflect that genome size data from O. delavayi (n=1) was removed from the manuscript here and throughout the manuscript.
- Line 130: O. armatus (n = 3 genotypes), O. fortunei (n = 4) 130 genotypes, O. fragrans (n = 12 genotypes, and O. heterophyllus (n = 14 geno-131 types). Text: The sum of these numbers is 33 genotypes. The name of species must be written as italic.
This was changed to the correct number (27) here and throughout the manuscript.
- Line 323: “By 72 hours post-pollination, mean pollen tube lengths 323 were 1.6 ± 0.39 mm, 0.99 ± 0.95 mm, and 0.067 ± 0.070 mm for interspecific, 324 intraspecific, and self-crosses, respectively.” Please, check again these numbers. How did you calculate these numbers?
These numbers are derived from the same data as Table 7. These numbers are the mean of all of the pollen tube lengths for each cross type (interspecific, intraspecific, and self) at each time point.
For example, there are four interspecific crosses listed in Table 7. Each cross had 3 to 5 flowers visualized for pollen tube growth. So the mean for all interspecific crosses at 72 hours is the mean of all flowers collected for the four interspecific crosses (between 12 and 20 flowers total).
This result is not meant to be reflected in any table, it is simply a statement to let the reader know the “grand mean” for each time point, for each cross type without adding another table to the manuscript.
- Line 327: Replace: 72 Text: 48
Thank you for the thorough review. Changed as suggested.
- Line 327: (Table 6)
Added as suggested.
- Line 358: reference(s)
The following references were added as citation for mating barriers:
Morimoto, T.; Inaoka, M.; Banno, K.; Itai, A. Genetic mapping of a locus controlling the intergeneric hybridization barrier between apple and pear. Tree Genet. Genomes 2020, 16, 5; DOI: 10.1007/s11295-019-1397-7.
Städler, T.; Florez-Rueda, A.M.; Roth, M. A revival of effective ploidy: the asymmetry of parental roles in endosperm-based hybridization barriers. Curr. Opin. Plant Biol. 2021, 61, 102015; DOI:10.1016/j.pbi.2021.102015.
- Line 439: You should add more references which are relevant for the present research.
Thirteen additional references have been added to the manuscript (three were added in the second round of review, in addition to the ten references added in the first round). These provide citations for self-incompatibility systems in general and in olive, mechanisms of SI systems, comparisons of floral morphology to related species, and general information on mating barriers.

Round 3
Reviewer 3 Report
Dear author,
Revised version of the manuscript is significantly improved. I have only one comment which is incorporated in the pdf of the revised version of the manuscript. Because that, I suggest minor revision.
I look forward to seeing your published paper.
Sincerely yours

Author Response
Thank you for your careful reviews of this manuscript. I agree that it is much improved since the original submission, and I look forward to reading the other papers in this Special Issue.
In line 117, "eight" was replaced with "seven" to reflect the removal of O. delavayi. The numbers now add to 33.